# Study on the Modification of Dietary Fiber and Degradation of Zearalenone in Corn Germ Meal by Solid-State Fermentation with *Bacillus subtilis* K6

**DOI:** 10.3390/foods14152680

**Published:** 2025-07-30

**Authors:** Jiahao Li, Kailong Li, Langwen Tang, Chun Hua, Na Chen, Chenxian Yang, Ying Xin, Fusheng Chen

**Affiliations:** College of Food Science and Technology, Henan University of Technology, Zhengzhou 450001, China; ljh20006211924@163.com (J.L.); lkail819@163.com (K.L.); langwen196@163.com (L.T.); huac327@163.com (C.H.); chenna2793@126.com (N.C.); yangcx@haut.edu.cn (C.Y.)

**Keywords:** corn germ meal, soluble dietary fiber, zearalenone degradation, solid-state fermentation, physicochemical properties, microstructural characterization

## Abstract

Although corn germ meal is a rich source of dietary fiber, it contains a relatively low proportion of soluble dietary fiber (SDF) and is frequently contaminated with high levels of zearalenone (ZEN). Solid-state fermentation has the dual effects of modifying dietary fiber (DF) and degrading mycotoxins. This study optimized the solid-state fermentation process of corn germ meal using *Bacillus subtilis* K6 through response surface methodology (RSM) to enhance SDF yield while efficiently degrading ZEN. Results indicated that fermentation solid-to-liquid ratio and time had greater impacts on SDF yield and ZEN degradation rate than fermentation temperature. The optimal conditions were determined as temperature 36.5 °C, time 65 h, and solid-to-liquid ratio 1:0.82 (*w*/*v*). Under these conditions, the ZEN degradation rate reached 96.27 ± 0.53%, while the SDF yield increased from 9.47 ± 0.68% to 20.11 ± 1.87% (optimizing the SDF/DF ratio from 1:7 to 1:3). Scanning electron microscopy (SEM) and confocal laser scanning microscope (CLSM) revealed the structural transformation of dietary fiber from smooth to loose and porous forms. This structural modification resulted in a significant improvement in the physicochemical properties of dietary fiber, with water-holding capacity (WHC), oil-holding capacity (OHC), and water-swelling capacity (WSC) increasing by 34.8%, 16.4%, and 15.2%, respectively. Additionally, the protein and total phenolic contents increased by 23.0% and 82.61%, respectively. This research has achieved efficient detoxification and dietary fiber modification of corn germ meal, significantly enhancing the resource utilization rate of corn by-products and providing technical and theoretical support for industrial production applications.

## 1. Introduction

Corn germ meal is the main by-product of corn germ oil extraction, with enormous annual production in China [1], making it a potential dietary fiber source. Studies have shown that the DF content in corn germ meal is as high as 55%, indicating its potential to be developed as a high-quality dietary fiber resource [2]. Consuming SDF is associated with a variety of health benefits, such as lowering blood lipid levels, improving blood sugar control, and reducing inflammation [3,4,5]. However, the soluble dietary fiber content in corn germ meal is less than 10. Additionally, zearalenone is commonly enriched in the meal during corn germ processing [6,7]. Therefore, it is necessary to develop a green processing technology that can simultaneously increase SDF content and efficiently degrade ZEN.

Microbial solid-state fermentation can not only remove harmful substances such as mycotoxins from raw materials but also improve the physicochemical structure of raw materials and enhance their biological activity [8,9,10,11,12]. At present, microbial solid-state fermentation has been widely used in DF modification and ZEN degradation [13,14,15]. For example, Chu, et al. [16] used *Bacillus natto* to ferment millet, increasing the SDF content from 2.3% to 13.2%, a nearly fivefold increase. Similarly, after solid-state fermentation of corn bran, the water-holding capacity, swelling capacity, and oil-binding capacity of SDF increased to 1.57 times, 1.95 times, and 1.80 times of those before fermentation, respectively [15]. Regarding the degradation rate of ZEN, Wang, et al. [17] evaluated the degradation capability of *Bacillus pumilus* ES-21 against ZEN in the culture medium, and found that the degradation rate of ZEN by the strain ES- reached up to 95.7%. Researchers have also conducted solid-state fermentation of wheat bran using *Aspergillus niger* and *Lactobacillus plantarum*, resulting in reductions of aflatoxin B1, deoxynivalenol, and ZEN contents by 76.34%, 18.67%, and 34.48%, respectively [18]. These studies demonstrate that solid-state fermentation combines the dual effects of dietary fiber modification and mycotoxin degradation, providing a feasible pathway for the safe and value-added utilization of corn germ meal.

This study has opened up a new approach for the application of *Bacillus subtilis* K6 (CCTCC M 20231096) in the fermentation of corn germ meal. This strain has the dual functions of cellulose biotransformation and ZEN bioremediation. An optimized solid-state fermentation system was established by using the response surface method, which has a unique synergistic effect of SDF enrichment and mycotoxin detoxification. Under the optimal conditions, the changes in the content of components such as SDF and insoluble dietary fiber (IDF) in corn germ meal were further analyzed, and the changes in the physicochemical indicators and microstructure of dietary fiber before and after fermentation were evaluated. By integrating process optimization with multi-parameter quality assessment, this study provides a transformative green processing platform for the safe and high-value utilization of by-products from grain processing.

## 2. Materials and Methods

### 2.1. Materials and Reagents

Corn germ meal (Sanxing Corn Industry Technology Co., Ltd., Jinan, China); corn alkenone Enzyme-linked Immunosorbent Assay Quantitative Detection Kit (Shanghai Yulong Biotechnology Co., Ltd., Shanghai, China); corn-enone immunoaffinity column (Pribolab Bioengineering Co., Ltd., Zouping, China); anhydrous ethanol and sulfuric acid (analytical grade) (Tianli Chemical Reagent Co., Ltd., Tianjin, China); β -mercaptoethanol, methyl red, bromocresol green (Analytical Grade) (Comeio Chemical Reagent Co., Ltd., Tianjin, China); alkaline protease, α -amylase, and amyloglucosidase (biological reagents) (Sola Biotechnology Co., Ltd., Beijing, China).

The bacterial strain *Bacillus subtilis* K6 (preservation number: CCTCC NO: M 20231096), previously isolated and metabolically optimized in our laboratory, was deposited at the CCTCC, Wuhan, China.

### 2.2. Sample Pretreatment

Foreign matter was removed from corn germ meal (stored in −20 °C refrigerator), followed by drying treatment. The dried material was subsequently pulverized and sieved through a 40-mesh sieve [19]. The resulting corn germ meal powder was collected and stored in a desiccator for subsequent experimental use.

### 2.3. Determination of ZEN Content in Corn Germ

A mass of 4.0 g of the sample was weighed into a 250 mL conical flask. Subsequently, 100 mL of methanol aqueous solution (methanol:water = 7:3, *w*/*v*) was added. The mixture was vortexed for 5 min and allowed to stand for 10 min. Following sedimentation, 1 mL of the supernatant was combined with 1 mL of purified water to prepare the test solution. The ZEN content was quantitatively analyzed by enzyme-linked immunosorbent assay [20].

### 2.4. Preparation of Fermentation Broth

Prepare fermentation broth by referring to the method of Shi, et al. [21]. A 100 μL aliquot of *Bacillus subtilis* K6 was aseptically transferred from cryovials into LB liquid medium and incubated at 37 °C with 180 rpm agitation for 12 h using a constant-temperature incubator shaker (HZQ-X3000, Shanghai Yiheng Scientific Instruments Co., Ltd., Shanghai, China) to obtain seed culture. The seed culture was subsequently inoculated at 5% *v*/*v* into sterile LB liquid medium and further incubated under identical conditions (37 °C, 180 rpm) for 12 h to prepare the fermentation broth.

### 2.5. Extraction of Dietary Fiber from Corn Germ Meal

Dietary fiber was extracted from corn germ meal by referring to and slightly modifying the method of Chen, et al. [22]. Samples were initially dried in an oven at 60 °C for 12 h. Subsequently, the dried samples were mixed with distilled water at a 1:15 *w*/*v* ratio, and the pH was adjusted to 6.5. Following the addition of 1.0% α-amylase, the mixture was subjected to continuous agitation at 90 °C for 1 h using a digital-controlled thermostatic magnetic stirring water bath (Model FJS-6, Changzhou Dingxin Experimental Instrument Co., Ltd., Changzhou, China). After cooling to 60 °C, the pH was adjusted to 4.5, and 1.0% glucoamylase was added for further hydrolysis under 60 °C water bath agitation for 1 h. The pH was then readjusted to 9.0, followed by the addition of 1.0% alkaline protease and incubation at 60 °C with agitation for 1 h. Enzymatic inactivation was achieved by boiling in a water bath for 5 min. After cooling to room temperature, the mixture was centrifuged to separate IDF as the precipitate. The supernatant was combined with four volumes of 95% ethanol preheated to 60 °C and allowed to stand for 24 h. Following centrifugation at 5000 rpm for 5 min, the resultant precipitate was freeze-dried using a vacuum freeze dryer (Model LGJ-25, Beijing Sihuan Scientific Instrument Factory, Beijing, China) to obtain SDF. At room temperature, the extracted SDF and IDF were stored in a desiccator for subsequent experiments (within one week).

### 2.6. Effect of Filling Amount on Solid-State Fermentation Efficiency

Referring to Bi, et al. [23], corn germ meal was divided into equal parts (5, 10, 15, 20, and 25 g) and respectively placed in 100 mL Erlenmeyer flasks (corresponding to packing densities of 0.05, 0.10, 0.15, 0.20, and 0.25 g/mL, respectively), then sterilized at 121 °C for 20 min. Fermentation was conducted in a constant temperature and humidity incubator (LHS-HC-I, Shanghai Yiheng Scientific Instruments Co., Ltd., Shanghai, China) at 37 °C for 60 h under the following conditions: NaHCO_3_ supplementation at 32 mg/g and a solid-to-liquid ratio of 1:0.8 *w*/*v*. Fermentation efficacy was evaluated based on total viable bacterial count and residual moisture content post-fermentation.

### 2.7. Effect of Amount of NaHCO_3_ Added on Solid-State Fermentation Efficiency

The original pH of corn germ meal was acidic, which was not conducive to the growth of the strain. Therefore, referring to da Silva, et al. [24] et al., NaHCO_3_ was added to adjust the pH. Corn germ meal (15 g) was loaded into 100 mL Erlenmeyer flasks and sterilized at 121 °C for 20 min. Varying concentrations of NaHCO_3_ (16, 32, 48, 64, and 80 mg/g) were supplemented under a solid-to-liquid ratio of 1:0.8 *w*/*v*, followed by fermentation at 37 °C for 60 h. Fermentation efficiency was evaluated based on total viable bacterial count.

### 2.8. Single-Factor Experiments on the Effects of Solid-State Fermentation on DF Yield and ZEN Degradation Rate

#### 2.8.1. Fermentation Temperature

An appropriate amount of corn germ meal was loaded into 100 mL Erlenmeyer flasks and sterilized at 121 °C for 20 min. Following supplementation with 2500 ppb ZEN and an optimal NaHCO_3_ concentration (*w*/*w*), the fermentation broth was inoculated at 8% (*v*/*w*) inoculum size. The solid-to-liquid ratio was adjusted to 1:0.8 *w*/*v*, and fermentation was conducted at varying temperatures (27 °C, 32 °C, 37 °C, 42 °C, and 47 °C) for 60 h. The SDF yield and ZEN degradation rate were quantitatively determined post-fermentation.

#### 2.8.2. Fermentation Time

Corn germ meal was aseptically loaded into 100 mL Erlenmeyer flasks at appropriate quantities and sterilized at 121 °C for 20 min. After supplementation with 2500 ppb ZEN and an optimized NaHCO_3_ concentration (*w*/*w*), the material was inoculated with 8% (*v*/*w*) fermentation broth. The solid-to-liquid ratio was adjusted to 1:0.8 *w*/*v*, followed by fermentation at 37 °C for varying durations (36, 48, 60, 72, and 84 h). The SDF yield and ZEN degradation rate were quantitatively determined post-fermentation.

#### 2.8.3. Solid-to-Liquid Ratio

Corn germ meal was loaded into 100 mL Erlenmeyer flasks at appropriate quantities and sterilized at 121 °C for 20 min. After supplementation with 2500 ppb ZEN and optimized NaHCO_3_ concentration (*w*/*w*), the material was inoculated with 8% *v*/*w* fermentation broth. The solid-to-liquid ratios were adjusted to 1:0.4, 1:0.6, 1:0.8, 1:1.0, and 1:1.2 *w*/*v*, followed by fermentation at 37 °C for 60 h. Post-fermentation analyses included determination of SDF yield and ZEN degradation rate.

### 2.9. Response Surface Optimization

Fermentation temperature (A), fermentation time (B), and solid-to-liquid ratio (C) were selected as experimental variables, with SDF yield (Y1) and ZEN degradation efficiency (Y2) serving as response parameters. A three-factor three-level RSM based on Box–Behnken design was implemented using Design-Expert 11.0 software (Trial version, Stat-Ease Incorporated, Minneapolis, MN, USA) to optimize solid-state fermentation conditions for corn germ meal. The experimental matrix comprised 17 systematic combinations generated by the BBD algorithm. Post-optimization validation experiments were conducted to confirm the predicted optimal parameters. Statistical significance of model terms was evaluated through analysis of variance, while response surface plots were generated to visualize factor interactions.

The derived second-order polynomial equations for response variables are:Y1 = 95.55 + 0.33A + 0.0179B + 3.98C + 0.00083AB − 0.24AC + 0.0288BC − 2.22A2 − 0.00378B2 − 5.16C2Y2 = 18.64 + 0.29A + 0.0196B + 0.34C − 0.00542AB + 0.02AC − 0.0188BC − 0.93A2 − 0.00264B2 − 2.04C2

### 2.10. Component Analysis of Corn Germ Meal

Analysis of cellulose (GB 5009.88-2023) [25], hemicellulose (GB 5009.88-2023) [25], acid-soluble lignin (GB/T 35818-2018) [26], acid-insoluble lignin (GB/T 35818-2018) [26], protein (GB 5009.5-2025) [27], ash content (GB 5009.4-2016) [28], and total phenolic substances (LS/T 6119-2017) [29] was carried out using the Chinese national standard method. The SDF and IDF contents in the corn germ meal were quantified according to the AOAC Official Method 991.43.

### 2.11. Determination of Physicochemical Properties of Dietary Fiber

#### 2.11.1. Water-Holding Capacity

The WHC was determined according to the method described by Shang, et al. [30] with modifications. Specifically, 1.00 g of freeze-dried sample was thoroughly mixed with 30 mL of distilled water and incubated at room temperature for 18 h. The mixture was centrifuged at 3000 r/min for 20 min, after which the aqueous phase was discarded. The remaining precipitate was weighed to calculate WHC using the following formula:$$WHC\;(g/g)\;=\;\frac{W_{1}-W_{0}}{W_{0}}$$
where *W*_1_ represents the mass (g) of the centrifuged precipitate after discarding the aqueous phase and *W*_0_ denotes the dry mass (g) of the original sample.

#### 2.11.2. Oil-Holding Capacity

The OHC was determined according to the method described by [31] with modifications. Specifically, 1.00 g of freeze-dried sample was weighed and thoroughly mixed with 25 mL of soybean oil. The mixture was incubated at room temperature for 18 h, followed by centrifugation at 3000 r/min for 20 min. The oil phase was discarded, and residual oil was removed using filter paper. The remaining precipitate was weighed to calculate OHC using the following formula:$$OHC\;(g/g)\;=\;\frac{M_{1}-M_{0}}{M_{0}}$$
where *M*_1_ represents the mass (g) of the sample after residual oil removal and *M*_0_ denotes the initial mass (g) of the sample.

#### 2.11.3. Water-Swelling Capacity

The WSC was determined according to the method described by Liao, et al. [32] with modifications. Briefly, 0.20 g of sample was weighed into a 10 mL graduated cylinder, followed by the addition of distilled water to the 10 mL mark. The mixture was vortexed thoroughly and allowed to stand at room temperature for 18 h. The final volume of the hydrated sample was recorded. The WSC was calculated using the following formula:$$WSC\;(mL/g)\;=\;\frac{V_{1}-V_{0}}{M}$$
where *V*_1_ represents the sample volume (mL) after hydration; *V*_0_ denotes the sample volume (mL) before hydration; and *M* corresponds to the mass (g) of the sample.

### 2.12. Observation of Dietary Fiber by CLSM

The dietary fiber and protein in the samples were stained with fluorescein isothiocyanate (FITC) and calcofluor white. The specific methods are as follows: Accurately weigh 10 mg of IDF and SDF before fermentation and those after fermentation, and place them respectively in 1.5 mL centrifuge tubes to prepare a 10 mg/mL mixed solution. In each sample, 20 µL of 0.1% FITC and 20 µL of 0.1% calcofluor white stain were added, respectively. After vortex for 1 min to mix evenly and staining for 15 min, 40 µL of the stained sample solution was taken and dropped onto a slide. After covering with a cover slip, it was tested on the machine.

The fluorescence parameters were as follows: calcofluor white (Ex/Em = 405/430 nm, indigo emission) and fluorescein isothiocyanate (Ex/Em = 480/517 nm, green emission).

### 2.13. Observation of Dietary Fiber by SEM

The microstructure of fermented dietary fiber was analyzed using a secondary electron detector and a scanning electron microscope (Quanta 250FEG, FEI Company, Hillsboro, WA, USA) [33]. Samples were affixed to metal stubs with conductive adhesive and sputter-coated with gold for 60 s. Scanning electron microscopy imaging was performed on gold-coated specimens at an acceleration voltage of 10 kv under low vacuum mode. Micrographs were captured at magnifications ranging from 500× to 5000× to characterize surface morphology and evaluate the microstructural features of the samples.

### 2.14. Statistical Analyses

All experiments were conducted in triplicate. The means were compared using one-way analysis of variance followed by Duncan’s test (*p* < 0.05) in SPSS 27.0 software to determine the significance of main effects. Data are expressed as mean ± standard deviation. Figures were plotted using Origin 2024 software (OriginLab Corporation, Northampton, MA, USA).

## 3. Results and Discussion

### 3.1. Effect of Filling Amount and Amount of NaHCO_3_ Added on Solid-State Fermentation Efficiency

The filling amount directly affects the growth environment and metabolic efficiency of the strain [34,35]. As corn germ meal is acidic and unfavorable for strain growth, it is essential to determine the optimal filling amount and pH [36]. The impact of filling amount on fermentation efficacy is illustrated in Figure 1A. Increasing the substrate loading caused the total bacterial count of *Bacillus subtilis* K6 to first rise and then decline. Concurrently, the moisture content of fermented corn germ meal showed a progressive increase during the filling process, stabilizing only when the filling amount surpassed 15 g. At a filling amount of 10 g, the maximum colony count reached 125.5 × 10^8^ CFU/g, with a post-fermentation moisture content of 59.83%. When the filling amount is 15 g, there is no difference in colony count, but the moisture content reaches up to 69.87% at the highest. This occurs because excessively low filling quantities accelerate moisture loss from corn germ meal, hindering strain proliferation. Selecting an appropriate filling amount regulates moisture dissipation rate, thereby facilitating fermentation. When the substrate loading exceeded 15 g, colony counts begin to decline, while the moisture content tends to stabilize. The primary reason is that excessive filling amount leads to compromised aeration of the culture medium, inhibiting strain growth. Concurrently, excessive heat generation and inadequate heat dissipation trigger microbial cell autolysis, ultimately reducing the total bacterial count [37,38]. Consequently, a filling amount of 15 g was selected for subsequent experiments based on comprehensive analysis.

The effect of amount of NaHCO_3_ added on fermentation efficacy is shown in Figure 1B. As the NaHCO_3_ dosage increased, the total bacterial count of *Bacillus subtilis* K6 exhibited an initial increase followed by a decline. At a NaHCO_3_ dosage of 32 mg/g, the maximum colony count was achieved. This phenomenon arises because corn germ meal itself is acidic, with an initial pH of approximately 3.0, which inhibits strain growth [39]. The addition of NaHCO_3_ elevates the initial pH of the substrate, not only improving the fermentation environment but also inducing cellulose expansion, thereby enhancing SDF yield [40,41]. When the NaHCO_3_ dosage exceeded 32 mg/g, the colony counts begin to decline, primarily due to the alkaline fermentation environment (pH > 7.0), which exceeded the strain’s tolerance threshold and suppressed its growth. Moreover, some of the literature shows that under acidic conditions, *Bacillus subtilis* has a poor degradation effect on ZEN, but when the pH is neutral or alkaline, it has a better degradation effect on ZEN [12]. Based on these findings, a NaHCO_3_ dosage of 32 mg/g was selected for subsequent experiments.

### 3.2. Solid-State Fermentation Single-Factor Experiments

#### 3.2.1. Effect of Fermentation Temperature on SDF Yield and ZEN Degradation Rate

The impact of fermentation temperature on ZEN degradation rate and SDF yield in corn germ meal is illustrated in Figure 2A. As the fermentation temperature increased, both SDF yield and ZEN degradation rate initially rose and subsequently declined. At 37 °C, both parameters reached their maximum values of 18.41% and 96.48%, respectively. Under lower or higher temperature conditions, both SDF yield and ZEN degradation rate were reduced. This is primarily due to inhibited strain growth at lower temperatures, whereas at higher temperatures, strain degeneration occurred alongside suppression of active substances secreted by the strain [42]. These findings align with reports by Chang [12] observed similar temperature-dependent ZEN degradation patterns using *Bacillus mojavensis* L-4 in corn germ meal fermentation. Based on comprehensive analysis, a temperature range of 32–42 °C was selected for subsequent RSM optimization experiments.

#### 3.2.2. Effect of Fermentation Time on SDF Yield and ZEN Degradation Rate

The effect of fermentation time on SDF yield and ZEN degradation rate in corn germ meal is shown in Figure 2B. As fermentation time increased, the SDF yield first rose and then declined, while the ZEN degradation rate initially increased and subsequently stabilized. At 60 h, both parameters reached their maximum values of 18.37% and 95.47%, respectively. During the early fermentation stage, insufficient duration led to incomplete fermentation and poor enzymatic hydrolysis, resulting in low SDF yield and ZEN degradation rate [12,43]. When the fermentation time exceeded 60 h, the degradation efficiency of ZEN tended to stabilize. This might be because as the fermentation time extended, the strain gradually adapted to the system environment and proliferated in large quantities. In the later stage of fermentation, the strains began to die out, resulting in no significant increase in the degradation rate of ZEN and it started to stabilize. In the study by Xiang et al., the curve of the degradation of ZEN by *Bacillus subtilis* over time also showed a similar trend [44]. Concurrently, the strain enters the logarithmic growth phase, during which nutrient depletion in the medium triggers DF consumption by the strain, consequently diminishing SDF yield [43,45]. Consequently, a fermentation time range of 48–72 h was selected for RSM experiments.

#### 3.2.3. Effect of Solid-to-Liquid Ratio on SDF Yield and ZEN Degradation Rate

The effect of fermentation solid-to-liquid ratio on SDF yield and ZEN degradation rate in corn germ meal is shown in Figure 2C. As the solid-to-liquid ratio increased, both SDF yield and ZEN degradation rate first increased and then decreased, reaching maximum values of 18.31% and 94.74%, respectively, at a ratio of 1:0.8. When the moisture content in the substrate was too low, strain growth was inhibited, and degradative active substances secreted by the strain could not fully contact the substrate. When the solid-to-liquid ratio exceeded 1:0.8, both SDF yield and ZEN degradation rate began to decline. This phenomenon may be attributed to the dilution of active metabolites secreted by the microbial strain, which reduces the concentration of bioactive compounds and consequently leads to diminished fermentation efficiency. It has also been proposed by researchers that excessive moisture content decreases substrate porosity and gas permeability, thereby impairing oxygen transfer capacity and CO_2_ removal efficiency, which ultimately creates unfavorable conditions for mycelial growth [46]. Therefore, after comprehensive consideration, a solid-to-liquid ratio range of 1:0.7–1:0.9 was selected for RSM experiments.

### 3.3. Response Surface Optimization Experiments

Based on the single-factor experiments, three factors-fermentation temperature, fermentation time, and solid-to-liquid ratio were selected as independent variables, with SDF yield and ZEN degradation rate designated as response values. The experiments were designed using the Box–Behnken method, and the corresponding factor level coding table is shown in Table 1.

#### 3.3.1. Response Surface Experimental Results and Regression Model Analysis

Using fermentation temperature (A), fermentation time (B), and solid-to-liquid ratio (C) as independent variables, with ZEN degradation rate (Y1) and SDF yield (Y2) as response variables, a three-factor three-level Box–Behnken experimental design was established. A total of 17 experimental groups with different combinations were tested for response values, and the results are listed in Table 2. The data were processed and analyzed using Design-Expert statistical software, generating regression models and ANOVA tables. The multivariate regression equations were derived as follows:Y1 = 95.55 + 0.33A + 0.0179B + 3.98C + 0.00083AB − 0.24AC + 0.0288BC − 2.22A2 − 0.00378B2 − 5.16C2Y2 = 18.64 + 0.29A + 0.0196B + 0.34C − 0.00542AB + 0.02AC − 0.0188BC − 0.93A2 − 0.00264B2 − 2.04C2

As shown in Table 3, the model exhibited a *p*-value < 0.001, while the lack-of-fit term (*p* = 0.0718 > 0.05) was non-significant, indicating that the regression model is highly significant. The correlation coefficient (R^2^ = 0.9843) and adjusted R^2^ (R^2^_Adj_ = 0.9642) demonstrated high model reliability, confirming its strong alignment with experimental outcomes, which validates its capability to accurately predict the effect of *Bacillus subtilis* K6 on SDF yield in corn germ meal. All selected factors significantly or highly significantly influenced SDF yield. The interaction between factors B and C was significant, while the quadratic terms A^2^, B^2^, and C^2^ exhibited highly significant effects on SDF yield. Furthermore, the influence magnitude of each factor on SDF yield was determined by their F-test values, where a higher F value corresponds to a stronger impact. Based on the F values in Table 3, the three factors affected ZEN degradation rate in the order of B > C > A (fermentation time > solid-to-liquid ratio > fermentation temperature).

The ANOVA results for this model are shown in the table below. As presented in Table 4, the model exhibited a *p*-value < 0.001, while the lack-of-fit term (*p* = 0.1133 > 0.05) was not statistically significant, indicating that the regression model is highly significant. The correlation coefficient (R^2^ = 0.9943) and adjusted R^2^ (R^2^_Adj_ = 0.9870) demonstrated high model reliability, confirming its strong alignment with experimental conditions. This validates the model’s capability to accurately predict the degradation efficacy of *Bacillus subtilis* K6 on ZEN in corn germ meal. The selected time and solid-to-liquid ratio in the model both exerted statistically important or highly important effects (*p* < 0.05 or *p* < 0.01) on the efficacy of *Bacillus subtilis* K6 in degrading ZEN. This is probably because the complex nature of the solid-state fermentation matrix rendered the strain’s ZEN-degrading activity less sensitive to temperature fluctuations. The interaction between factors B (time) and C (solid-to-liquid ratio) was significant, while the quadratic terms A^2^ (temperature), B^2^, and C^2^ showed statistically highly important effects on ZEN degradation. Furthermore, the influence magnitude of each factor on ZEN degradation was determined by their F-test values, where a higher F value indicates a stronger impact. Based on the F values in Table 4, the three factors affected ZEN degradation rate in the order of C > B > A (solid-to-liquid ratio > fermentation time > fermentation temperature).

#### 3.3.2. Response Surface Analysis

Based on the regression models, response surface plots for SDF yield and ZEN degradation rate were generated as shown in Figure 3. Analysis of these plots revealed the interaction strength between factors: steeper response surfaces and elliptical-shaped contour lines indicate significant interactions between paired factors, whereas flatter surfaces and circular contours denote non-significant interactions [47]. Figure 3A1,A2,C1 and C2 illustrate the interaction intensity between the solid-to-liquid ratio and the other two factors. When the solid-to-liquid ratio ranged from 1:0.8–1:0.85, both ZEN degradation rate and SDF yield reached their maxima during *Bacillus subtilis* K6 fermentation of corn germ meal, with a significant interaction observed between the solid-to-liquid ratio and fermentation time. Figure 3A1,A2,B1,B2 depict the interaction intensity between fermentation time and the other two factors. When fermentation time fell within 57.6 h–67.2 h, ZEN degradation rate and SDF yield peaked, demonstrating a significant interaction between fermentation time and solid-to-liquid ratio. Figure 3B1,B2,C1 and C2 reflect interactions between fermentation temperature and the other two factors. Within the temperature range of 36–38 °C, *Bacillus subtilis* K6 fermentation achieved the highest ZEN degradation rate and SDF yield in corn germ meal.

#### 3.3.3. Determination and Verification of the Optimal Fermentation Process

Based on the response surface analysis, the optimal fermentation conditions for balancing SDF yield and ZEN degradation rate were determined as follows: fermentation temperature of 36.502 °C, fermentation time of 65.208 h, and solid-to-liquid ratio of 1:0.8167. Under these predicted parameters, the SDF yield and ZEN degradation rate were projected to reach 18.493% and 95.844%, respectively. To validate the model, verification experiments were conducted under practical conditions: 36.5 °C, 65 h, and 1:0.82. The actual SDF yield (20.11 ± 1.87%) and ZEN degradation rate (96.27 ± 0.53%) closely matched the predicted values, demonstrating the model’s precision and reliability in forecasting SDF yield and ZEN degradation during *Bacillus subtilis* K6 fermentation of corn germ meal.

### 3.4. Microstructure Observation of Dietary Fiber

The microstructure of dietary fiber before and after fermentation was observed by CLSM and SEM. It can be known from Figure 4 that before fermentation, IDF presented a reticular structure and SDF presented a sheet-like structure, with clear and complete structures. After fermentation, the structure of IDF collapses and no longer has a clear framework structure, while the sheet-like structure of SDF is damaged. This change increases the specific surface area of DF, which in turn increases the contact area between DF and lipids, thus enhancing OHC. The surface microstructure of IDF and SDF before and after fermentation was further observed by SEM. As shown in the Figure 5, pre-fermentation, IDF exhibited a smooth surface devoid of visible pores, while SDF displayed irregular flake-like structures with dense surfaces. Post-fermentation, the surface structure of IDF became fragmented, forming irregular porous configurations, whereas SDF transformed into a loose, rough texture with numerous microvoids, resembling a honeycomb-like structure and exhibiting increased specific surface area. Similar structural changes also occurred after Lin, et al. [13] fermented okara dietary fiber. After Ma, et al. [48] treated the rice bran dietary fiber with enzymes, a similar phenomenon also occurred. Their results show that the lumps and spherical substances on the surface of rice bran dietary fiber have almost disappeared, and the fibrous layered structure has been destroyed. This porous structure is due to the cellulase and hemicellulase secreted by the strain targeting the glycosidic bonds in cellulose and hemicellulose, decomposing insoluble polymers, transforming the originally smooth and dense fiber matrix into a loose and porous structure, exposing hydrophilic groups, and directly enhancing the WHC and WSC of dietary fibers [10,13,48,49]. The partial cleavage and dissolution of glycosidic bonds further contributed to the formation of these irregular and rough morphological features [50].

### 3.5. Changes in Components of Corn Germ Meal Before and After Fermentation

The compositional changes in corn germ meal before and after solid-state fermentation with *Bacillus subtilis* K6 are summarized in Table 5. Post-fermentation, the SDF content in fermented corn germ meal significantly increased (*p* < 0.05) from 9.47% to 19.55%, representing a 2.06-fold enhancement. Conversely, the IDF content decreased significantly (*p* < 0.05) from 55.76% to 45.31% [51]. Some researchers have also discovered similar phenomena [13]. This phenomenon may result from the enzymatic hydrolysis of IDF by *Bacillus subtilis* K6, which converts IDF into soluble low-molecular-weight compounds, thereby elevating SDF levels [52]. Consequently, the SDF/IDF ratio improved from 1:7 to 1:3. Furthermore, cellulose, hemicellulose, and acid-insoluble lignin contents in corn germ meal all showed significant reductions (*p* < 0.05), with cellulose declining most markedly from 30.30% to 22.70%, suggesting that insoluble cellulose degradation primarily contributed to the SDF increase. This aligns with the observed rise in SDF content from 9.47% to 19.55%. The total phenol content also increased significantly (*p* < 0.05), which might be due to *Bacillus subtilis* K6 destroying the structure of IDF and releasing the phenolic substances previously embedded in the IDF matrix, which was consistent with the results of SEM and CLSM [50,53]. As phenolic compounds typically exhibit strong antioxidant properties, the elevated total phenolics imply enhanced antioxidant potential in the fermented dietary fiber [54,55,56].

### 3.6. Effects of Fermentation on Physicochemical Properties of Dietary Fiber

The effects of fermentation modification on the water-holding capacity (WHC), oil-holding capacity (OHC), and water-swelling capacity (WSC) of dietary fiber (DF) are summarized in Table 6. Dietary fiber with good hydration properties can promote intestinal peristalsis and reduce the risk of intestinal diseases and obesity [57,58]. After fermentation modification, WHC significantly increased (*p* < 0.05) from 8.40 g/g to 11.32 g/g, indicating enhanced hydrophilicity of DF. This improvement is likely due to fermentation-induced cleavage of bonds within the fiber, altering its surface structure to expose hydrophilic groups that form hydrogen bonds with water molecules, thereby increasing water-binding capacity [48,49,59]. OHC improved (*p* < 0.05) from 11.99 g/g to 13.96 g/g, suggesting that fermentation disrupted glycosidic bonds in polysaccharide chains, exposing more lipophilic groups and significantly enhancing oil retention [60]. WSC increased (*p* < 0.05) from 7.06 mL/g to 8.13 mL/g, possibly due to structural changes in the fiber, including reduced particle size and increased specific surface area, which expanded fiber-water contact and improved water absorption [32,61]. These are consistent with the results of SEM and CLSM. The structure of dietary fiber after fermentation becomes loose, exposing the internal groups and increasing the specific surface area, which enhances WHC, OHC, and WSC. Additionally, the elevated proportion of SDF contributed to the enhanced WHC and WSC performance [61]. Ting et al. modified the dietary fiber in tea residue through fermentation, and the results showed that its WHC, WSC, and OHC increased by 31%, 73%, and 52% respectively [57].

### 3.7. Inference on the Mechanism of DF Modification and ZEN Degradation by Fermentation

During the solid-state fermentation process, with the total dietary fiber content remaining unchanged, the yield of soluble dietary fiber increased from 9.47% to 19.55%, and the SDF/IDF ratio also significantly improved. This might be due to the strain secreting cellulase and hemicellulase targeting the glycosidic bonds in cellulose and hemicellulose, which cleave insoluble polymers into soluble oligosaccharides [10,49]. Structural analysis indicates that the originally smooth and dense fibrous matrix transforms into a loose and porous structure, exposing hydrophilic groups (hydroxyl and carboxyl groups), which directly enhances the hydration capacity [48,49]. This might be the main reason for the conversion of IDF to SDF. The significant increase in WHC, OHC, and WSC of dietary fiber after fermentation also confirms this point.

After fermentation, zearalenone in corn germ meal is almost completely degraded. Zearalenone undergoes microbial degradation when microorganisms secrete their metabolites or enzymes during their growth and development. This might be because *Bacillus subtilis* K6 can convert the structure of ZEN into α- and β-zearalenol under the action of enzymes, and finally lyse its cyclic structure, which greatly reduces its toxicity [44,62].

## 4. Conclusions

In this study, solid-state fermentation of corn germ meal using *Bacillus subtilis* K6 was conducted to simultaneously achieve dietary fiber modification and ZEN degradation. Because corn germ meal is acidic and has a high fiber content, it is prone to absorbing water and swelling. Therefore, the addition of amount of NaHCO_3_ and filling amount was optimized. The experimental results show that the optimal filling amount is 15 g and the optimal amount of NaHCO_3_ added is 32 mg/g. Under this condition, response surface optimization is carried out. The results demonstrated that the optimal process conditions were as follows: fermentation temperature of 36.5 °C, fermentation time of 65 h, and solid-to-liquid ratio of 1:0.82 (g/mL). Under these conditions, the SDF yield increased from 9.47 ± 0.68% to 20.11 ± 1.87%, representing a 3.04-fold enhancement compared to pre-fermentation levels. The ZEN degradation rate reached 96.27 ± 0.53%.

CLSM and SEM revealed that the dietary fiber structure transitioned from a smooth to a loose and porous morphology post-fermentation. The physicochemical properties of dietary fiber were also significantly improved after solid-state fermentation. Water-holding capacity, oil-holding capacity, and water-swelling capacity increased by 34.8%, 16.4%, and 15.2%, respectively. This might be because the structure of dietary fiber becomes loose after fermentation, exposing internal groups and increasing the specific surface area. These are consistent with the results of scanning electron microscopy. Furthermore, protein, ash, and total phenolic contents in corn germ meal were significantly elevated post-fermentation, and the SDF/DF ratio was optimized from 1:7 to 1:3.

The solid-state fermentation process of corn germ meal with *Bacillus subtilis* K6 established in this study has greatly improved the utilization rate of resources. Compared with traditional liquid fermentation, it requires relatively less water and is a sustainable and environmentally friendly technology that improves resource utilization. In addition, compared with other methods, this technology can achieve simultaneous detoxification and fiber transformation, and it is efficient and economical. Admittedly, this experiment still has certain limitations, such as in the biological safety assessment of fermented products and other aspects. However, some studies have shown that biological detoxification is effective, specific, and safe [63,64]. Chen, et al. [65] also demonstrated that ZEN in feed can be effectively degraded through Bacillus fermentation, demonstrating good safety. In the future, we will also continue to go further and directly verify the safety of this method, and study its potential and value in the application of high-fiber baked goods, meat products, and instant beverage food systems. This strategy might be applicable to modify other food ingredients with high fiber content, creating a new source of high dietary fiber raw materials. It also could potentially be applied to other food by-products, such as soybean meal and rice bran, to degrade mycotoxins and enhance food safety. This method not only provides a technically and economically feasible solution for the high-value and safe utilization of corn processing by-products, but also offers reference and guidance for the degradation of mycotoxins and the modification of dietary fiber on other raw materials.

## Figures and Tables

**Figure 1 foods-14-02680-f001:**
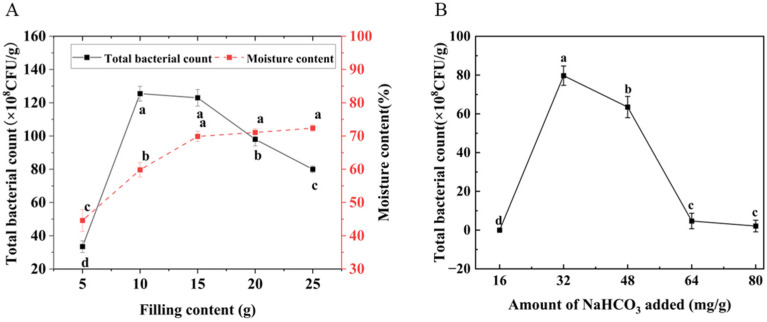
Effect of filling amount and amount of NaHCO_3_ added on total bacterial count. (**A**) The influence of filling amount on total bacterial count and moisture content. (**B**) The influence of NaHCO_3_ on total bacterial count. Note: Different lowercase letters indicate significant differences (*p* < 0.05).

**Figure 2 foods-14-02680-f002:**
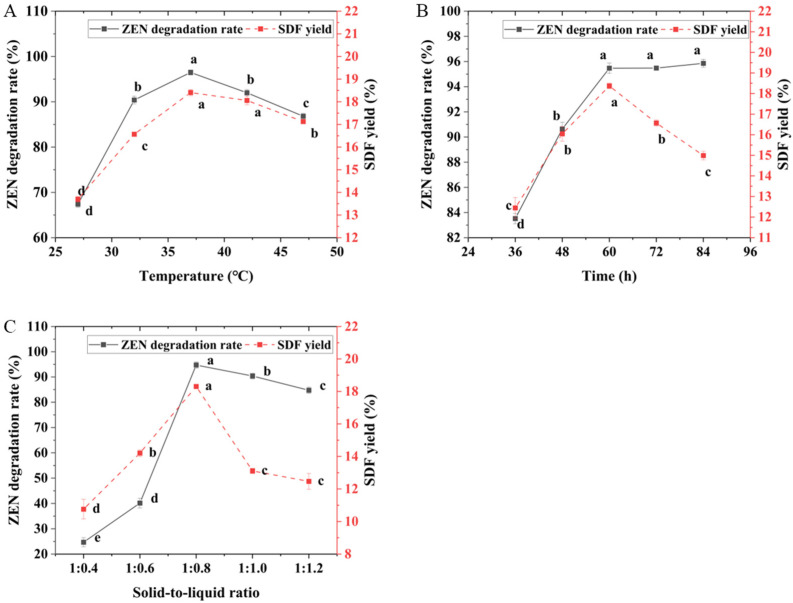
Influence of different factors on SDF yield and ZEN degradation rate. (**A**) Fermentation time, (**B**) fermentation temperature, (**C**) solid-to-liquid ratio. Note: Different lowercase letters indicate significant differences (*p* < 0.05).

**Figure 3 foods-14-02680-f003:**
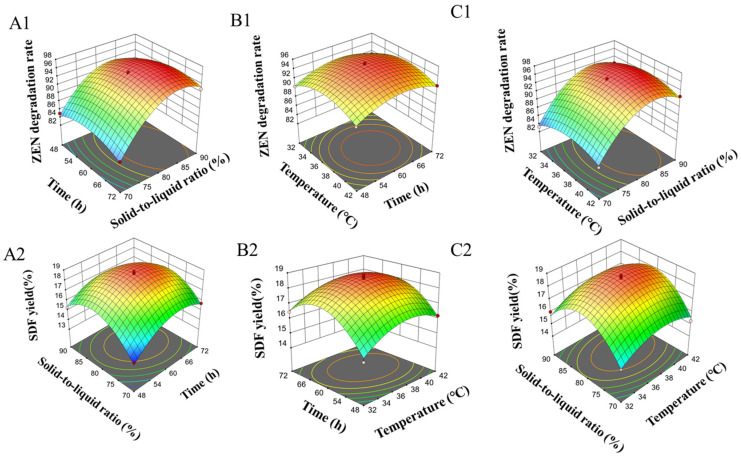
Response surface curve graph of fermented corn germ meal. (**A1**) Influence of time and solid-to-liquid ratio on degradation rate of ZEN, (**B1**) influence of time and temperature on degradation rate of ZEN, (**C1**) influence of temperature and solid-to-liquid ratio on degradation rate of ZEN (**A2**) influence of time and solid-to-liquid ratio on yield of SDF, (**B2**) influence of time and temperature on yield of SDF, (**C2**) influence of temperature and solid-to-liquid ratio on yield of SDF.

**Figure 4 foods-14-02680-f004:**
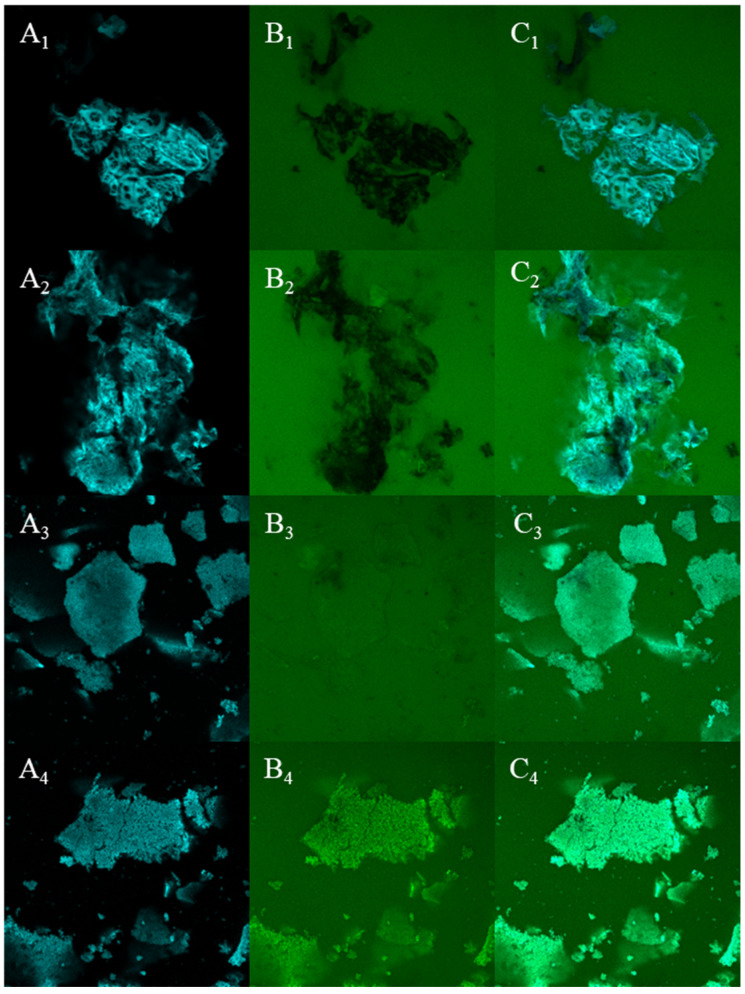
CLSM microstructure observation of IDF and SDF before and after fermentation. (**A_1_**) IDF fiber staining, (**A_2_**) FIDF fiber staining, (**A_3_**) SDF fiber staining, (**A_4_**) FSDF fiber staining, (**B_1_**) IDF protein staining, (**B_2_**) FIDF protein staining, (**B_3_**) SDF protein staining, (**B_4_**) FSDF protein staining, (**C_1_**) IDF combination diagram, (**C_2_**) FIDF combination diagram, (**C_3_**) SDF combination diagram, (**C_4_**) FSDF combination diagram (50 µm).

**Figure 5 foods-14-02680-f005:**
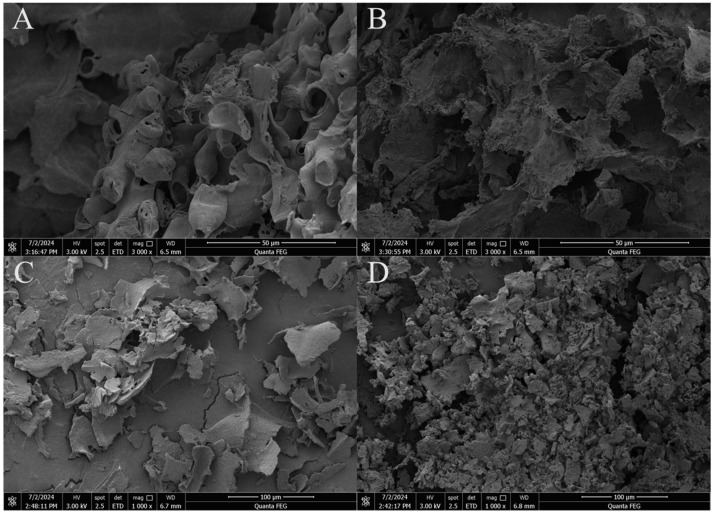
SEM microstructure observation of IDF and SDF before and after fermentation. (**A**) IDF (3000×), (**B**) FIDF (3000×), (**C**) SDF (1000×), (**D**) FSDF (1000×).

**Table 1 foods-14-02680-t001:** Response surface factor level coding table.

Coded Level	Factors		
	Temperature (°C)	Time (h)	Solid-to-Liquid Ratio (g/mL)
−1	32	48	1:0.7
0	37	60	1:0.8
1	42	72	1:0.9

**Table 2 foods-14-02680-t002:** Response surface design and results.

Number	A	B	C	Y1 (%)	Y2 (%)
1	42	72	1:0.8	92.05	17.08
2	37	60	1:0.8	95.89	18.83
3	37	72	1:0.9	92.81	15.18
4	32	48	1:0.8	90.3	15.05
5	37	60	1:0.8	95.55	18.65
6	42	60	1:0.9	92.64	16.35
7	37	60	1:0.8	95.48	18.42
8	37	60	1:0.8	95.79	18.51
9	37	60	1:0.8	95.04	18.79
10	42	48	1:0.8	90.89	16.20
11	32	60	1:0.7	83.23	15.04
12	42	60	1:0.7	84.40	15.28
13	32	72	1:0.8	91.37	16.45
14	32	60	1:0.9	92.42	16.02
15	37	48	1:0.9	90.83	15.33
16	37	72	1:0.7	84.23	15.73
17	37	48	1:0.7	84.99	14.08

**Table 3 foods-14-02680-t003:** Regression model and analysis of variance for SDF yield.

Source of Variation	Sum of Squares	df	Mean Square	F-Value	*p*-Value	Significance
Model	38.36	9	4.26	48.84	<0.0001	**
A: Temperature	0.6903	1	0.6903	7.91	0.0261	*
B: Time	1.79	1	1.79	20.47	0.0027	**
C: Solid-to-liquid ratio	0.9453	1	0.9453	10.83	0.0133	*
AB	0.0676	1	0.0676	0.7746	0.4080	
AC	0.0020	1	0.0020	0.0232	0.8832	
BC	0.8100	1	0.8100	9.28	0.0187	*
A^2^	3.61	1	3.61	41.39	0.0004	**
B^2^	9.71	1	9.71	111.29	<0.0001	**
C^2^	17.54	1	17.54	201.04	<0.0001	**
Residual	0.6109	7	0.0873			
Lack of Fit	0.4869	3	0.1623	5.24	0.0718	Not significant
Pure Error	0.1240	4	0.0310			
Total	38.97	16				

Note: ** *p* < 0.01 (highly significant); * *p* < 0.05 (significant).

**Table 4 foods-14-02680-t004:** Regression model and analysis of variance for ZEN degradation rate.

Source of Variation	Sum of Squares	df	Mean Square	F-Value	*p*-Value	Significance
Model	297.84	9	33.09	136.26	<0.0001	**
A: Temperature	0.8845	1	0.8845	3.64	0.0980	
B: Time	1.49	1	1.49	6.13	0.0425	*
C: Solid-to-liquid ratio	126.80	1	126.80	522.12	<0.0001	**
AB	0.0020	1	0.0020	0.0083	0.9298	
AC	0.2256	1	0.2256	0.9290	0.3672	
BC	1.88	1	1.88	7.73	0.0273	*
A^2^	20.75	1	20.75	85.44	<0.0001	**
B^2^	19.96	1	19.96	82.20	<0.0001	**
C^2^	112.00	1	112.00	461.17	<0.0001	**
Residual	1.70	7	0.2429			
Lack of Fit	1.26	3	0.4206	3.84	0.1133	Not significant
Pure Error	0.4382	4	0.1095			
Total	299.54	16				

Note: ** *p* < 0.01 (highly significant); * *p* < 0.05 (significant).

**Table 5 foods-14-02680-t005:** Changes in composition of corn germ meal before and after fermentation.

Component	Before Fermentation	After Fermentation
IDF (%)	55.76 ± 0.74 ^a^	45.31 ± 1.06 ^b^
SDF (%)	9.47 ± 0.68 ^a^	19.55 ± 0.83 ^b^
Cellulose (%)	30.30 ± 0.10 ^a^	22.70 ± 1.00 ^b^
Hemicellulose (%)	26.00 ± 0.50 ^a^	24.00 ± 0.35 ^b^
Acid-Soluble Lignin (%)	9.30 ± 0.10 ^a^	9.40 ± 0.05 ^a^
Acid-Insoluble Lignin (%)	7.00 ± 0.20 ^a^	5.50 ± 0.20 ^b^
Protein (%)	16.80 ± 1.36 ^a^	20.67 ± 0.97 ^b^
Ash (%)	2.88 ± 0.04 ^a^	3.75 ± 0.02 ^b^
Total Phenolics (%)	6.50 ± 0.11 ^a^	11.87 ± 0.31 ^b^

Note: Different lowercase letters indicate significant differences (*p* < 0.05).

**Table 6 foods-14-02680-t006:** Effects of fermentation on physicochemical properties of dietary fiber.

	DF	FDF
WHC (g/g)	8.40 ± 0.09 ^a^	11.32 ± 0.07 ^b^
OHC (g/g)	11.99 ± 0.16 ^a^	13.96 ± 0.13 ^b^
WSC (mL/g)	7.06 ± 0.05 ^a^	8.13 ± 0.10 ^b^

Note: Different lowercase letters indicate significant differences (*p* < 0.05).

## Data Availability

In this work, all relevant information and techniques are offered. The corresponding author should be contacted with any further questions.

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
