# Peer review of "Study on the Modification of Dietary Fiber and Degradation of Zearalenone in Corn Germ Meal by Solid-State Fermentation with Bacillus subtilis K6"

_foods, 2025, doi:10.3390/foods14152680_

Round 1
Reviewer 1 Report
Comments and Suggestions for Authors
The manuscript investigates the solid-state fermentation process applied to corn germ meal using Bacillus subtilis K6, with the aim of increasing the soluble fiber content and degrading the mycotoxin zearalenone. The authors demonstrated that, under optimal conditions, a significant increase in soluble fiber content and an efficient degradation of zearalenone were achieved.
My observations are as follows:
I believe that the phrase “Effect of filling amount on solid-state fermentation efficiency” is not sufficiently clear, even though the volume of the Erlenmeyer flask is specified in the methods section.
As the authors themselves mention, fermentative processes and biochemical transformations are influenced by the pH of the medium. Therefore, it would have been more appropriate to report the results in relation to pH rather than solely based on the quantity of NaHCO₃ added.
The Results section indicates optimal solid-state fermentation conditions as follows: solid-to-liquid ratio of 1:0.8–1:0.85, fermentation time between 57.6–67.2 h, and temperature in the range of 36–38°C. In the Conclusions section, the optimal conditions are listed as 36.5°C, fermentation time of 65 h, and I believe these should be revised. This is especially important considering that, earlier in the Methods section, it was mentioned that fermentation was carried out at 37°C for 60 h.
Additionally, greater attention should be given to the figure numbering, both in the captions and within the main body of the text.
Author Response
请参阅附件。

Reviewer 2 Report
Comments and Suggestions for Authors
1. The end of the summary could benefit from a better conclusion, for example, with its practical or industrial application.
2. It is recommended that some keywords be different from those in the title. In addition, use more standardized or indexable terms.
3. The introduction could be improved by reducing repetitions and emphasizing the scientific novelty of the study.
4. In section 2.1, include the brand, city, and country of all materials and reagents in parentheses.
5. All equipment should have the model, brand, city, and country in parentheses.
6. Sections 2.2, 2.4, 2.5, 2.6, and 2.7 do not have methodological sources.
7. In section 2.7. (line 133), correct NaHCO3.
8. The mathematical model used in the experimental design (regression equations) should be described in section 2.9, not just in the results. Clearly explain the statistical methods used to validate the RSM models.
9. The units of measurement should be specified more consistently (w/v, w/w, v/v, etc.).
10. There is no indication of how or for how long the samples were stored before the physicochemical and structural analyses.
11. In 2.10, the “Chinese national standard method” is mentioned, but the specific standard number should be included.
12. In 2.12 and 2.13, details such as laser type and wavelength for CLSM, and vacuum conditions, pressure, or detector type for SEM are missing.
13. In 2.4, include details of the software used, such as Design Expert v9.0 software (Trial version, Stat-Ease Inc., Minneapolis, MN, USA) and Origin Pro 2025 software (OriginLab Corporation, Northampton, MA, USA).
14. Avoid repeating data already presented in tables and figures; focus more on their interpretation.
15. Deepen the discussion of the mechanisms behind the increase in FDS and the degradation of ZEN.
16. Compare the results with more recent and relevant studies, not just with references from China.
17. Explain more clearly how the porous structure improves the functional properties of the product.
18. Emphasize more the relationship between the structural changes observed (CLSM and SEM) and the physicochemical properties.
19. Include limitations of the study, for example, whether or not the biosafety of the fermented product was evaluated.
20. Add comments on the technological applicability or industrial scalability of the results.
21. Avoid generic statements such as “significant improvement” without statistical or contextual support.
22. Specify whether there were statistically significant differences between treatments, using specific p-values when possible.
23. Check the consistency between the numerical data reported in the text and those presented in the tables.
24. In addition to making comparisons with other studies, the biological, physical, and chemical mechanisms involved in the results obtained should be explored.
25. Discuss whether the process is viable at the industrial level, considering aspects such as costs, scalability, and impact.
26. It would be advisable to include more recent scientific articles that support and enrich the discussion of the results.
27. The conclusions should be improved, including the limitations of the study and possible new lines of research. The potential for application in the food industry should also be highlighted.
28. It is recommended to reduce the similarity index of the iThenticate software (27%), especially in the methodology, results, and discussion sections.
Round 2
Reviewer 2 Report
Comments and Suggestions for Authors
Accept in the current form.